# Critique of enhanced power claimed for Quasi-ANCOVA and Dual-Centered ANCOVA

**B. Wade Brorsen** [1] *, **Hua Lin**[2], **Robert E. Larzelere**[2]

**1** Department of Agricultural Economics, Oklahoma State University, Stillwater, Oklahoma, United States of America, **2** Department of Human Development and Family Science, Oklahoma State University, Stillwater, Oklahoma, United States of America

* wade.brorsen@okstate.edu

## Abstract

Two related methods, Quasi-ANCOVA and Dual-Centered ANCOVA, have recently been suggested as a way to get greater power when analyzing data from a before and after study. Both methods use group-mean centering where the groups are the treatment and comparison groups. Group-mean centering creates a generated regressor problem. When the estimated standard errors are corrected for the generated regressor problem, there is no longer any gain in power. The corrected Quasi-ANCOVA estimates are identical to those from ANOVA and the corrected Dual-Centered ANCOVA estimates are identical to those from the differences model. These conclusions are derived analytically and also verified using a Monte Carlo simulation.

## Introduction

Two methods of analyzing data that have been suggested recently include Quasi-ANCOVA for randomized pre-post data [1] and Dual-Centered ANCOVA for longitudinal data over two occasions [2]. When using standard errors from ordinary least squares (OLS), these methods both appear to duplicate the corresponding estimated treatment effects of ANOVA and differences-in-differences, respectively, while generally increasing statistical power. This note argues that there is no increase in power from these methods once standard errors are corrected for the presence of a generated regressor. As Pagan [3] argued, in the presence of a generated regressor, the standard errors need to be calculated using the two-stage least squares (2SLS) formula. As noted by Greene [4] (p. 58) the key difference is that the 2SLS formula for the residual variance uses the actual value of the variable rather than the generated value. Once the 2SLS formula for the residual variance is applied, the estimated treatment effect of quasi-ANCOVA and its standard error are the same as with analysis of variance (ANOVA). Similarly, estimates and inference from Dual-Centered ANCOVA are the same as with the differences model. Thus, there is no gain in power from using either Quasi-ANCOVA or Dual-Centered ANCOVA once the correct standard errors are used.

**Data Availability Statement:** The analysis is based on analytical derivations and a Monte Carlo simulation and so does not use data. An example copy of the R program used for the Monte Carlo

simulation is included in the supplementary materials.

**Funding:** This research benefited from funding from NICHD grant #5 R03 HD107307 to Robert Larzelere and Hua Lin. Brorsen receives funding from the Oklahoma Agricultural Experiment Station, and United States Department of Agriculture National Institute of Food and Agriculture [Hatch Project number OKL03170]. The funders had no role in study design, analysis, decision to publish, or preparation of manuscript.

**Competing interests:** The authors have declared that no competing interests exist.

## Alternative models

First consider three basic models: (i) Analysis of Variance (ANOVA), (ii) differences, and (iii) Analysis of Covariance (ANCOVA). The ANOVA model is

$$Y_{ij1} = \beta_0 + \beta_1 X_j + \varepsilon_{ij} \tag{1}$$

where $Y_{ij1}$ is the posttest score (the subscript 1 indicates posttest) of the $i$th person that receives the $j$th treatment, $X_j$ is an indicator variable for the $j$th treatment ($X_2 = 1$ for those receiving the treatment and $X_1 = 0$ for the control group), and $\varepsilon_{ij}$ is the mean-zero error term.

Many behavioral studies are longitudinal and thus provide both a pretest and a posttest. One approach to analyze such data is difference analysis (it is a special case of differences in differences when the time series has length two). The differences model is

$$Y_{ij1} - Y_{ij0} = \delta_0 + \delta_1 X_j + \vartheta_{ij} \tag{2}$$

where $Y_{ij0}$ is the pretest score and $\vartheta_{ij}$ is a mean-zero error term that differs from $\varepsilon_{ij}$.

The ANCOVA model relaxes the differences model's assumption that the coefficient on the pretest score is one:

$$Y_{ij1} = \alpha_0 + \alpha_1 X_j + \alpha_2 Y_{ij0} + \nu_{ij} \tag{3}$$

where $\nu_{ij}$ is the ANCOVA error term. Huitema [1] suggests Quasi-ANCOVA as an alternative to ANCOVA in randomized studies when one or more covariate test scores are not obtained until after the treatment has begun. We limit the exposition herein to the case of a late pretest score.

## Quasi-ANCOVA

Now consider the generated regressor problem created by Quasi-ANCOVA [1]. Quasi-ANCOVA modifies (3) by subtracting the group means of the pretest score from each pretest score. Following Pagan's [3, p. 227] Model 2, Quasi-ANCOVA can be represented as a two-equation system (a recursive simultaneous equation system):

$$Y_{ij1} = \gamma_0 + \gamma_1 X_j + \gamma_2 (Y_{ij0} - \bar{Y}_{i.0}) + \nu_{ij} \tag{4}$$

$$Y_{ij0} = \varphi_0 + \varphi_1 X_j + \tau_{ij0} \tag{5}$$

where the group means of the pretest score $\bar{Y}_{i.0}$ are the predicted values of Eq (5) and $\tau_{ij0}$ is a random error term. Note that the same symbol is used for the error term in Eqs (3) and (4) since the error terms are equal. Since Quasi-ANCOVA uses the predictions of the second equation in calculating an explanatory variable, it creates a generated regressor problem. Note that the term $(Y_{ij0} - \bar{Y}_{i.0})$ is orthogonal to the treatment effect $X_j$ by construction (the term $Y_{ij0} - \bar{Y}_{i.0}$ is equal to the error term of (5) and with OLS, the explanatory variable $X_j$ and the error are assumed uncorrelated). Therefore, the estimated treatment effect with Quasi-ANCOVA will be the same as with ANOVA ($\hat{\beta}_1 = \hat{\gamma}_1$). As Pagan [3] argues, the standard errors for the estimate of $\gamma_1$ in Eq (4) will be too small if calculated using the OLS formula. Instead, standard errors need to be calculated using the two-stage least squares formula (this is the same formula as with instrumental variables). In the 2SLS formula, the standard errors in (4) are calculated using the actual value of $Y_{ij0}$ rather than its predicted value ($\bar{Y}_{i.0}$). Letting $\bar{Y}_{i.0} = Y_{ij0}$ in (4) gives $Y_{ij0} - \bar{Y}_{i.0} = Y_{ij0} - Y_{ij0} = 0$, and so the 2SLS standard deviation of the error in (4) is exactly the same as the OLS standard deviation from ANOVA in Eq (1). Thus,

the estimated treatment effect and its standard error are exactly the same in Quasi-ANCOVA as in ANOVA.

Note that Pagan [3] argues that generated regressor problems can also be solved by using full information maximum likelihood (FIML). That does not work with Quasi-ANCOVA as the parameters are not all identified (the parameter $\gamma_2$ cannot be estimated with FIML since Eq (5) has no exogenous variable that is not also in Eq (4) and so the order condition for identification is not met).

An alternative way to think about Quasi-ANCOVA is as a seemingly unrelated regression [4, p. 328]. Note that the added regressor in (4) is the same as the residual in (5): $(Y_{ij0} - \bar{Y}_{i.0} = \hat{\tau}_{ij})$. Therefore, the Quasi-ANCOVA model can also be represented as

$$Y_{ij1} = \gamma_0 + \gamma_1 X_j + \varepsilon_{ij} \tag{6}$$

$$Y_{ij0} = \varphi_0 + \varphi_1 X_j + \tau_{ij0}. \tag{7}$$

where $cov(\varepsilon_{ij}, \tau_{ij0}) \neq 0$. From (4), the Quasi-ANCOVA assumes $\varepsilon_{ij} = \gamma_2 \tau_{ij0} + v_{ij}$ and so $cov(\varepsilon_{ij}, \tau_{ij}) = cov(\gamma_2 \tau_{ij} + v_{ij}, \tau_{ij})$. Since the errors in (6) and (7) are correlated, it would appear that there could be a potential gain in efficiency from using seemingly unrelated regressions. However, since both equations have the same explanatory variables, there is no gain to using seemingly unrelated regressions in this case. That is, $\varepsilon_{ij}$ represents the residuals after regressing $Y_{ij1}$ on $X_j$. Eq (4) merely divides those residuals into two parts: a portion that covaries with the residuals in regressing $Y_{ij0}$ on $X_j$ and a remaining portion. Thus, Eqs (6) and (7) provide an alternative argument to reach the same conclusion: Quasi-ANCOVA and ANOVA produce the same estimates of coefficients and standard errors when the Quasi-ANCOVA standard errors are calculated correctly.

## Dual-Centered ANCOVA

Dual-Centered ANCOVA as defined by Lin & Larzelere [2] is the same as Quasi-ANCOVA except the pretest score is subtracted from the left side:

$$Y_{ij1} - \bar{Y}_{i.0} = \omega_0 + \omega_1 X_j + \gamma_2 (Y_{ij0} - \bar{Y}_{i.0}) + v_{ij} \tag{8}$$

$$Y_{ij0} = \varphi_0 + \varphi_1 X_j + \tau_{ij0}. \tag{9}$$

As before, $(Y_{ij0} - \bar{Y}_{i.0})$ is orthogonal to $X_j$ by construction. As a result, the treatment effect with Dual-Centered ANCOVA is the same as the differences model in (2). The 2SLS standard error is also the same as the standard error in (2). Note that $\bar{Y}_{i.0}$ must be replaced by $Y_{ij0}$ on both the left side and the right side of Eq (8) when calculating the standard errors for the estimate of $\omega_1$. Thus, there is also no gain in using Dual-Centered ANCOVA once the standard errors are corrected for the presence of a generated regressor.

## Monte Carlo study

A Monte Carlo study in Table 1 is included to verify the analytical results. The simulation is based on the null hypothesis of a randomized pre-post design, in which the treatment effects are unbiased for Eqs (2) and (3). The simulation notation uses Eq (3) with the intercept set at 130 although its value is immaterial, the treatment effect ($\delta$ in Eq (2) and $\alpha_1$ in Eq (3)) is set at zero in all simulations, and varying values of $\alpha_2$ are used. The simulation included 1,000 replications.

**Table 1. Monte Carlo study of alternative treatment effects models.**

| model | True $\alpha_2$ | Mean of OLS Estimates $\widehat{\alpha}_2$ | $se(\widehat{\alpha}_2)$ | $\widehat{\alpha}_1$ | $se(\widehat{\alpha}_1)$ | MC SD $sd(\widehat{\alpha}_1)$ | MSE |
|---|---|---|---|---|---|---|---|
| ANOVA | 0.9 | - | - | -0.0002 | 0.948 | 0.998 | 224.8 |
| ANCOVA | 0.9 | 0.9 | 0.01 | -0.0068 | 0.413 | 0.411 | 42.7 |
| Quasi-ANCOVA | 0.9 | 0.9 | 0.01 | -0.0002 | 0.413 | 0.998 | 42.7 |
| Differences | 0.9 | - | - | -0.0074 | 0.424 | 0.418 | 44.9 |
| Dual-Centered | 0.9 | 0.9 | 0.01 | -0.0074 | 0.413 | 0.418 | 42.7 |
| ANOVA | 0.5 | - | - | -0.0052 | 0.948 | 0.990 | 224.8 |
| ANCOVA | 0.5 | 0.5 | 0.03 | -0.0121 | 0.822 | 0.831 | 168.6 |
| Quasi-ANCOVA | 0.5 | 0.5 | 0.03 | -0.0052 | 0.821 | 0.990 | 168.6 |
| Differences | 0.5 | - | - | -0.0165 | 0.948 | 0.934 | 224.6 |
| Dual-Centered | 0.5 | 0.5 | 0.03 | -0.0165 | 0.821 | 0.934 | 168.6 |
| ANOVA | -0.9 | - | - | -0.0153 | 0.948 | 0.942 | 224.7 |
| ANCOVA | -0.9 | -0.9 | 0.01 | -0.0003 | 0.413 | 0.435 | 42.7 |
| Quasi-ANCOVA | -0.9 | -0.9 | 0.01 | -0.0153 | 0.413 | 0.942 | 42.7 |
| Differences | -0.9 | - | - | -0.0321 | 1.847 | 1.821 | 853.7 |
| Dual-Centered | -0.9 | -0.9 | 0.01 | -0.0321 | 0.413 | 1.821 | 42.7 |

Note: The Monte Carlo study was conducted for the null hypothesis, i.e., a zero true treatment effect (means of pretest and posttest for all treatments were set at 130 with $SD = 15$). MSE is the mean squared error from the simulation and so its expected value is 225 for the ANOVA model.

Table 1 presents selected results of the Monte Carlo study. The key column of interest is the actual standard deviation of the estimated treatment effect $(sd(\widehat{\alpha}_1))$. This column shows that the standard deviation of the estimated treatment effects for ANOVA and Quasi-ANCOVA are identical. Further, the standard deviation of the estimated treatment effects of Differences and Dual-Centered ANCOVA are also exactly the same. Thus, the Monte Carlo results in Table 1 exactly match the theory.

Note that Quasi-ANCOVA produces the treatment effects of the ANOVA model, yet it uses the residual standard deviation and the treatment effect standard error of the ANCOVA model. Similarly, dual-centered ANCOVA produces the treatment effects of the differences model, but uses the residual standard deviation and treatment effect standard error of the ANCOVA model. All of the methods produce unbiased estimates of the treatment effects under the stated assumptions. The problem with Quasi-ANCOVA and Dual-Centered ANCOVA is in the calculation of the standard error.

## Interaction effects

As in Lin and Larzelere [2], an interaction term can be added to the dual-centered ANCOVA model in (8), which provides

$$Y_{ij1} - \bar{Y}_{i.0} = \omega_0 + \omega_1 X_j + \gamma_2(Y_{ij0} - \bar{Y}_{i.0}) + \gamma_3 X_j(Y_{ij0} - \bar{Y}_{i.0}) + v_{ij} \qquad (10)$$

We know from previous arguments that the coefficient on the treatment variable will be the same as in the differences model since the two added terms are orthogonal. We also know that 2SLS standard errors (where all three $\bar{Y}_{i.0}$ are replaced by $Y_{ij0}$ in the calculation) will need to be used for $\omega_1$. OLS standard errors can be used for estimates of $\gamma_2$ and $\gamma_3$ [3]. Now consider the

ANCOVA model with an interaction term

$$Y_{ij1} = \alpha_0 + \alpha_1 X_j + \alpha_2 Y_{ij0} + \alpha_3 X_j Y_{ij0} + \nu_{ij}. \tag{11}$$

From algebra (and verified with Monte Carlo simulation), $\alpha_2 = \gamma_2 + 1$ and $\alpha_3 = \gamma_3$. So, Dual-Centered ANCOVA provides the treatment effect of the differences model and the interaction effect from the ANCOVA model in (11), including the correct standard error for the interaction effect, but not for the treatment effect.

## Discussion

Quasi-ANCOVA and dual-centered ANCOVA do not have increased power once the standard errors are corrected for the statistical problem of using generated regressors. In fact, quasi-ANCOVA is identical to ANOVA and dual-centered ANCOVA is identical to the differences model. Thus, the claim of enhanced power for these two techniques does not hold.

Given that group-mean centering can cause the standard errors reported by standard statistical packages to be incorrect, it raises the question: when does group-mean centering create a generated regressor problem? Group-mean centering is often used to disaggregate within-group fixed effects and between-group fixed effects, but can create unforeseen problems [5]. The first point that we want to make is that with both quasi-ANCOVA and dual-centered ANCOVA, the group used for the group mean is the treatment condition, which creates an identification problem. Centering around individual means does not always create an identification problem and so FIML can work in some cases.

Further, note that grand-mean centering does not create any statistical problems. Grand-mean centering is a linear transformation of an explanatory variable and so it changes nothing other than the intercept.

Hill et al. [6, p. 124] discuss three different methods of using group-mean centering:

*The demeaned predictor variable (i.e., $x_{ij} - \bar{x}_j$) can then be included with the group mean $\bar{x}_j$, resulting in $y = a + B_1\bar{x}_j + B_2(x_{ij} - \bar{x}_j) + u$. Second, the group mean $\bar{x}_j$ can be included along with the unaltered, or raw, predictor $x_{ij}$, resulting in $y = a + B_1 x_{ij} + B_2\bar{x}_j + u$. Third is to simply use the demeaned value, resulting in $y = a + B_1(x_{ij} - \bar{x}_j) + u$.*

Pagan's [3] Model 4 directly addresses Hill's first method. He argues that OLS standard errors are asymptotically correct for $B_2$ (there could still be a benefit from a degrees of freedom adjustment in small samples), while 2SLS would be needed for $B_1$. With Hill's model two, it is $B_1$ that can use OLS standard errors and 2SLS standard errors are needed for $B_2$. As Pagan [3] discusses, OLS standard errors can be used for Hill's model three.

Bliese et al. [7, p. 81] suggests using random effects in the demeaned case because standard OLS packages will use an incorrect degrees of freedom. Using random effects does not solve the generated regressor problem.

Hoffman [8] also discusses baseline centering in longitudinal studies, where the values are centered around the initial value at the baseline time. Since the baseline value is not estimated, there is no generated regressor problem.

Zyphur et al. [9] consider the special case of a general cross-lagged model for panel data. In this case, group-mean centering can introduce dynamic panel bias. Dynamic panel bias is a different problem than generated regressors and so it is not addressed here.

Similarly, Rights and Sterba [5] consider the problem of random conflated slopes (slope heterogeneity) in Hill's Model 1 and Model 2. The problem that Rights and Sterba [5] consider is distinct from the generated regressor problem, but does indicate that fixed effects can be biased and standard errors incorrect in complex multilevel models.

As Pagan [3] (p. 283) says "the use of predictors or residuals does not necessarily lead to efficiency losses or incorrect standard errors" so it is hard to develop a general rule. But, in many cases, standard computer programs will not provide the correct standard errors when using group-mean centered data. The correct standard errors can usually be calculated using the 2SLS formula for standard errors or by estimating the equations jointly using full-information maximum likelihood.

## Supporting information

**S1 File. Example R code.**
(DOCX)

## Author Contributions

**Conceptualization:** Hua Lin, Robert E. Larzelere.

**Formal analysis:** B. Wade Brorsen.

**Funding acquisition:** Hua Lin, Robert E. Larzelere.

**Methodology:** B. Wade Brorsen.

**Project administration:** Robert E. Larzelere.

**Software:** Hua Lin.

**Validation:** B. Wade Brorsen, Hua Lin, Robert E. Larzelere.

**Writing – original draft:** B. Wade Brorsen.

**Writing – review & editing:** Hua Lin, Robert E. Larzelere.

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
