## [Decision Letter · Decision Letter 0]

13 Nov 2024

PONE-D-24-10379Critique of Enhanced Power Claimed for Quasi-ANCOVA and Dual-Centered ANCOVAPLOS ONE

Dear Dr. Brorsen,

Thank you for submitting your manuscript to PLOS ONE. After careful consideration, we feel that it has merit but does not fully meet PLOS ONE’s publication criteria as it currently stands. Therefore, we invite you to submit a revised version of the manuscript that addresses the points raised during the review process.

We look forward to receiving your revised manuscript.

Kind regards,

Surya Prakash Bhatt, Ph.D

Academic Editor

PLOS ONE

Journal Requirements:

4. Thank you for stating the following financial disclosure: “This research benefited from funding from NICHD grant #5 R03 HD107307 to Robert Larzelere and Hua Lin. Brorsen receives funding from the Oklahoma Agricultural Experiment Station, and United States Department of Agriculture National Institute of Food and Agriculture [Hatch Project number OKL03170].”

5. We note that your Data Availability Statement is currently as follows: “All relevant data are within the manuscript and in Supporting Information files.”

Please confirm at this time whether or not your submission contains all raw data required to replicate the results of your study. Authors must share the “minimal data set” for their submission. PLOS defines the minimal data set to consist of the data required to replicate all study findings reported in the article, as well as related metadata and methods (https://journals.plos.org/plosone/s/data-availability#loc-minimal-data-set-definition). For example, authors should submit the following data: - The values behind the means, standard deviations and other measures reported; - The values used to build graphs; - The points extracted from images for analysis. Authors do not need to submit their entire data set if only a portion of the data was used in the reported study. If your submission does not contain these data, please either upload them as Supporting Information files or deposit them to a stable, public repository and provide us with the relevant URLs, DOIs, or accession numbers. For a list of recommended repositories, please see https://journals.plos.org/plosone/s/recommended-repositories. If there are ethical or legal restrictions on sharing a de-identified data set, please explain them in detail (e.g., data contain potentially sensitive information, data are owned by a third-party organization, etc.) and who has imposed them (e.g., an ethics committee). Please also provide contact information for a data access committee, ethics committee, or other institutional body to which data requests may be sent. If data are owned by a third party, please indicate how others may request data access.

Reviewers' comments:

Reviewer's Responses to Questions

**Comments to the Author**

1. Is the manuscript technically sound, and do the data support the conclusions?

Reviewer #1: Yes

Reviewer #2: Partly

2. Has the statistical analysis been performed appropriately and rigorously? 

Reviewer #1: Yes

Reviewer #2: Yes

3. Have the authors made all data underlying the findings in their manuscript fully available?

Reviewer #1: Yes

Reviewer #2: Yes

4. Is the manuscript presented in an intelligible fashion and written in standard English?

Reviewer #1: Yes

Reviewer #2: Yes

5. Review Comments to the Author

Reviewer #1: I have not read the primary sources related to the method the authors critique. As such, I am assuming that they have faithfully described it.

This paper challenges the claim that the Quasi-ANOVA and dual-centered ANOVA models provide a power boost when assessing a treatment effect in a randomized controlled trial with pre-test data. The authors of this critique argue that both of these models underestimate the size of the standard error for the treatment effect because one of the predictors in the model is estimated, not observed, creating a generated regressor bias problem. Once this bias is addressed through 2SLS regression, the results of both of the above models converge with the results from standard models, producing no change in statistical power. I was fully convinced by these author's arguments and identified no errors in logic or the algebraic manipulation of equations.

A minor point: I was initially confused by the subscript 1 for the leftmost term in equation (1) until I got to equation (2). This confusion could be addressed when explaining the terms to equation (1).

Reviewer #2: The problem is simple. It is about pre-post test analysis. The ANOVA model is adequate. Two of the authors proposed a new model for the problem claiming better power. Now, in the current article, the authors proved that the power of both the models are the same. The original article was published in the Journal of adolescence. I suggest that the critique be submitted to the same journal.

6. PLOS authors have the option to publish the peer review history of their article (what does this mean?). If published, this will include your full peer review and any attached files.

Reviewer #1: **Yes: **David B. Wilson

Reviewer #2: No

---

## [Author Response · Author response to Decision Letter 0]

2 Dec 2024

The one comment of the reviewers was

A minor point: I was initially confused by the subscript 1 for the leftmost term in equation (1) until I got to equation (2). This confusion could be addressed when explaining the terms to equation (1).

This was addressed in the parentheses following equation (1) by explaining that the subscript meant posttest.

---

## [Editor Report · Decision Letter 1]

7 Jan 2025

Critique of Enhanced Power Claimed for Quasi-ANCOVA and Dual-Centered ANCOVA

PONE-D-24-10379R1

Dear Dr. Brorsen,

We’re pleased to inform you that your manuscript has been judged scientifically suitable for publication and will be formally accepted for publication once it meets all outstanding technical requirements.

Kind regards,

Surya Prakash Bhatt, Ph.D

Academic Editor

PLOS ONE
---

## [Editor Report · Acceptance letter]

12 Jan 2025

PONE-D-24-10379R1 

PLOS ONE

Dear Dr. Brorsen, 

I'm pleased to inform you that your manuscript has been deemed suitable for publication in PLOS ONE. Congratulations! Your manuscript is now being handed over to our production team.

Kind regards, 

on behalf of

Dr. Surya Prakash Bhatt 

Academic Editor

PLOS ONE